# Relative Excess Risk of Metabolic Syndrome Due to Interaction Between Handgrip Strength and Dietary Patterns Among Korean Youth

**DOI:** 10.3390/nu17142282

**Published:** 2025-07-10

**Authors:** Seong Woong Yoon, Hunju Lee, Hyowon Choi, Yunkoo Kang

**Affiliations:** 1Department of Medicine, Yonsei University Wonju College of Medicine, Wonju 26426, Republic of Korea; yoonsw0429@naver.com; 2Department of Prevention Medicine, Yonsei University Wonju College of Medicine, Wonju 26426, Republic of Korea; hjlee5371@gmail.com; 3Department of Pediatrics, Yonsei University Wonju College of Medicine, Wonju 26426, Republic of Korea

**Keywords:** adolescent, metabolic syndrome, muscle strength, feeding behavior

## Abstract

**Background/Objectives**: Metabolic syndrome (MetS) in adolescence increases chronic disease risk in adulthood. No study has explored the combined effects of skeletal muscle strength and dietary patterns in MetS. This study aimed to examine the individual and combined effects of dietary patterns and HGS on MetS and its components in Korean adolescents. **Methods**: Using the 2014–2019 Korea National Health and Nutrition Examination Survey data, a weighted sample of approximately 3.75 million adolescents was included. Dietary patterns were derived using principal component analysis. Relative handgrip strength (HGS) was calculated. Multivariable logistic regression and relative excess risk due to interaction (RERI) were used to assess dietary patterns, HGS, and MetS, stratified by sex and adjusted for age, smoking, alcohol consumption, economic status, residential area, and physical activity. **Results**: Low HGS was independently associated with a high odds of developing MetS in both men (OR, 1.108; 95% CI, 1.038–1.182) and women (OR, 1.128; 95% CI, 1.047–1.216). In contrast, dietary pattern alone was not significantly associated with MetS. Men with both low HGS and unhealthy dietary patterns (processed fat or Western diet) had higher odds of developing MetS, even though the interaction was sub-additive, as indicated by negative RERI values (processed fat: −0.22; Western diet: −0.11). **Conclusions**: Low HGS was a significant risk factor for MetS in Korean adolescents. Although no synergistic interaction was observed, low HGS remained a significant independent risk factor for MetS, underscoring the need to promote muscular strength in adolescents even in the absence of an unhealthy diet.

## 1. Introduction

Metabolic syndrome (MetS) in adolescence, which frequently persists into adulthood and increases the risk of type 2 diabetes and cardiovascular disease, has shown a rising prevalence among Korean adolescents [1,2,3,4,5,6,7,8,9]. The key pathophysiological mechanism underlying MetS is insulin resistance (IR), characterized by a reduced biological response of peripheral tissues to insulin [3,5,6,7]. In particular, adolescents are known to be more susceptible to IR due to increased secretion of growth hormone during puberty [10].

Skeletal muscle, the most abundant insulin-sensitive tissue, plays a central role in glucose uptake and storage, and contributes to insulin regulation through the secretion of various myokines and metabolites such as myostatin and irisin [3,5,6,11,12,13]. Handgrip strength (HGS) is a reliable and simple measure of skeletal muscle strength. Evidence from previous studies, including meta-analyses, suggests that lower HGS is associated with greater insulin resistance and, consequently, a higher risk of MetS in adolescents [3,4,5]. In Korean adolescents aged 10–18, Jung et al. [3] reported a significant P-for-trend for normalized combined HGS (CHGS/body weight) with both metabolic syndrome and insulin resistance after adjusting for sex, age, physical activity, and sedentary time. Wen et al. [5] demonstrated a linear dose–response relationship: each 0.1-unit increase in relative HGS (HGS/body weight) was associated with lower odds of MetS (OR 0.68, 95% CI 0.62–0.75) after adjustment for age, sex, BMI, socioeconomic status, smoking, and physical activity.

Agodi et al. reported that among adults aged 25 to 64 years, adherence to a prudent diet reduced the odds of MetS, while Western dietary adherence increased them [14]. Such diets are known to stimulate insulin secretion, promote fat storage, and significantly inhibit adipose tissue lipolysis and fatty acid oxidation [7,8,15]. Recent studies have also examined the relationship between dietary patterns and metabolic risk in adolescents [16,17].

Although previous studies have examined the relationship between dietary patterns and metabolic health in adolescents, few have investigated how skeletal muscle strength may modify this association. Given that insulin resistance can be influenced by both dietary intake and muscle mass, it is plausible that adolescents with lower muscle strength may be more vulnerable to the adverse metabolic effects of unhealthy eating patterns. Therefore, this study aimed to examine the individual and combined effects of dietary patterns—derived using principal component analysis (PCA)—and HGS on metabolic syndrome and its components in Korean adolescents. The analysis utilized nationally representative data from the Korea National Health and Nutrition Examination Survey (KNHANES) and considered potential sex-specific differences.

## 2. Materials and Methods

### 2.1. Data Source

This study analyzed data from the KNHANES, a yearly cross-sectional survey by the Korean Disease Control and Prevention Agency using a multistage cluster sampling design to assess the health and nutrition of a nationally representative Korean population [18]. The data for the selected participants had specific sampling weight factors because each datapoint did not have an equal probability of being selected. Data were analyzed after adjusting for specific sampling weight factors for each participant.

### 2.2. Study Population

Given that HGS was investigated only from 2014 to 2019, 3350 adolescents aged 12–18 years who participated during this period were included in this study. Adolescents with incomplete data of the following were excluded: sampling weights (*n* = 716), any component of MetS (*n* = 235); HGS or dietary pattern (*n* = 71); and other covariates, including age, smoking status, alcohol consumption, economic status, physical activity, and residential area (*n* = 171). Adolescents with energy intake <500 kcal and >5000 kcal/day were also excluded (*n* = 39). Ultimately, 2118 adolescents met the eligibility criteria (Figure 1).

### 2.3. Definition of MetS

MetS is defined according to the International Diabetes Federation consensus for children and adolescents as the presence of abdominal obesity and ≥2 additional clinical features [19]. Central obesity was determined as waist circumference ≥90th percentile or greater than or equal to the adult cut-offs (90 cm for men, 85 cm for women) for children aged 10–15 years, and adult cut-offs were used for adolescents aged 16–18 years. Additional clinical features included elevated triglyceride level (TG) (≥150 mg/dL), low high-density lipoprotein cholesterol (HDL-C) level (<40 mg/dL for both sexes aged 10–15 years and men aged 16–18 years; <50 mg/dL for women aged 16–18 years), hypertension (systolic blood pressure [SBP] ≥130 mmHg or diastolic blood pressure ≥85 mmHg), and impaired fasting glucose (IFG) level (≥100 mg/dL).

### 2.4. Assessment of HGS and Definition of Low Handgrip Strenght

HGS was measured in kilograms using a digital grip strength dynamometer (T.K.K. 5401; Takei, Niigata, Japan) in the standing position, with feet hip-width apart, arms naturally extended and hanging downward, and wrists in a neutral position. Grip strength was measured alternately for both hands, with 30 s rest intervals, and three measurements were taken per hand. The maximum value from each hand was summed to calculate the combined HGS (CHGS; kg). To account for potential bias due to body size, CHGS was normalized by body weight (nCHGS [%] = CHGS × 100/body weight [kg]). Low HGS was defined as nCHGS ≤ 10th percentile, stratified by sex and age.

### 2.5. Covariates

The covariates included sex, age, economic status, smoking status, alcohol consumption, residential area, and physical activity. Smoking status was categorized as either ever-smoker or never-smoker, based on self-reported use of any tobacco products, including conventional cigarettes and e-cigarettes. Alcohol consumption was defined as the consumption of at least one alcoholic beverage per month in the past year. Although the legal smoking and drinking age in Korea is ≥19 years, smoking experience and occasional alcohol use (≥1 drink/month) is reported among adolescents; therefore, we included smoking status and alcohol consumption as a covariate [20]. Economic status was classified into quartiles by dividing the household average monthly income by the square root of the number of household members. Residential areas were classified as either urban or rural. Physical activity was assessed based on whether participants met the recommended level of at least 1 h of moderate-to-vigorous physical activity per day that was sufficient to increase breathing and slightly elevate the heart rate [21].

### 2.6. Assessment of Dietary Patterns

Total food intake was assessed through home interviews using the 24 h dietary recall method conducted by trained interviewers, 1 week after the health interview and examination. Food items were classified into 29 food groups according to the Korean Nutrient Database to facilitate interpretation [22]. Owing to the high consumption of grains and their derivatives in Korea, this category was further subdivided into rice dishes, noodles and dumplings, bread and confectionery, and porridge and soups. Kimchi, a culturally important fermented food, was isolated from the vegetable group. Beverages were categorized as sugar-sweetened beverages, non-sugar-sweetened beverages, and alcoholic drinks.

Dietary patterns were identified using PCA based on the percentage contribution (g/day) of each food group. It was performed to reduce dimensionality and extract the underlying dietary structures of the participants. The first three principal components were retained based on eigenvalues, scree plot inspection, and interpretability [23]. Each participant was assigned a dietary pattern corresponding to the component with the highest score. The resulting patterns were labeled as Balanced, Processed fat, and Western, according to the major contributing food groups in each component. To support the interpretation of dietary patterns, visualizations of the PCA results were generated, including a loading heatmap and a biplot with 95% confidence ellipses by dietary group (Figure 2).

### 2.7. Statistical Analysis

All statistical analyses were performed using both weighted and unweighted methods. The participants were stratified by sex, and comparisons were made according to the presence or absence of MetS. Age was analyzed using Student *t*-test, and values are presented as mean ± standard deviation. Categorical variables were compared using the chi-square test and expressed as percentages (%). To illustrate the temporal trends in the prevalence of MetS and its components, the annual weighted prevalence estimates were calculated from 2014 to 2019, stratified by sex. These trends were visualized using line plots with 95% confidence intervals (CIs).

To examine the association between dietary patterns and HGS with metabolic components, design-corrected regression analysis was used [24]. Multivariate logistic regression analyses were conducted separately for male and female participants. All models were adjusted for age, smoking status, alcohol consumption, economic status, residential area, and physical activity level. The additive interaction between dietary patterns and low HGS on the risk of MetS was evaluated using the relative excess risk due to interaction (RERI) [25]. The RERI can be interpreted as the additional risk due to interactions and is calculated as the difference between the expected and observed risks.

All data were reconstructed and preprocessed using R version 4.3.2, (R Foundation for Statistical Computing, Vienna, Austria), with a *p*-value of <0.05 considered statistically significant.

## 3. Results

Table 1 presents the weighted baseline characteristics of Korean adolescents stratified by sex and the presence of MetS, including 1,959,239.78 men and 1,790,509.52 women. Among male participants, those with MetS were significantly older than those without MetS (16.2 ± 0.2 vs. 15.2 ± 0.1 years, *p* = 0.005). The overall distribution of dietary patterns did not differ significantly between the groups (*p* = 0.168). Low HGS was significantly more prevalent in the MetS group (36.1%) than in the non-MetS group (8.8%; *p* = 0.002). Other covariates, including economic status, residential area, smoking status, alcohol consumption, and physical activity, did not differ significantly between the groups. In female participants, those with MetS also had a higher mean age than those without MetS (16.4 ± 0.4 vs. 15.2 ± 0.1 years, *p* = 0.005). No significant differences were observed in dietary patterns (*p* = 0.288). Low HGS was observed in 66.1% of women with MetS compared to only 9.0% of those without MetS (*p* = 0.003). Economic status was also significantly associated with MetS (*p* = 0.018), with a greater proportion of participants with MetS belonging to the mid- and low-income groups. No significant differences were found in the residential area, smoking status, alcohol consumption, or physical activity.

Figure 3 shows the annual prevalence trends of MetS and its components among Korean adolescents from 2014 to 2019, stratified by sex. The prevalence of MetS and central obesity was consistently higher in men, with both increasing over time, and peaking in 2019. Hypertension in men declined until 2017 and rose thereafter, whereas that in women had a stable, lower trend. Triglyceride abnormalities fluctuated in both sexes, with a sharp increase in men and a decrease in women in 2019. Low HDL-C levels were more prevalent in women, illustrating a large year-to-year variability and a marked decrease in 2019.

Table 2 summarizes the associations among HGS, dietary patterns, and the odds of MetS and its components, stratified by sex. Adolescents with low HGS had significantly higher odds of developing MetS than those with normal HGS in both men (odds ratio [OR]: 1.108, 95% CI: 1.038–1.182) and women (OR: 1.128, 95% CI: 1.047–1.216). Low HGS was also significantly associated with higher odds of central obesity (men: OR 1.596, 95% CI 1.438–1.771; women: OR 1.395, 95% CI 1.258–1.547), low HDL cholesterol levels (men: OR 1.201, 95% CI 1.091–1.322; women: OR 1.177, 95% CI 1.061–1.306), and, in women only, hyperglycemia (OR: 1.156, 95% CI 1.060–1.260). In contrast, dietary patterns (processed fat or Western diet) were not significantly associated with higher odds of MetS or its components in either sex than the balanced dietary pattern. Results from unweighted models are presented in Appendix A.

Table 3 presents the interaction effects between HGS and dietary patterns on MetS and its components, using ORs and RERI with 95% CIs, stratified by sex. Among men, a combination of low HGS and processed fat diet was associated with higher odds of MetS (OR: 1.223, 95% CI: 1.055–1.418) and a sub-additive interaction (RERI: −0.22, 95% CI: −0.37 to −0.07). A similar pattern was observed for central obesity (OR: 1.771, 95% CI: 1.483–2.115; RERI: −0.75, 95% CI: −0.93 to −0.57). For hypertension and low HDL in men, the combination of low HGS and Western diet was associated with higher odds (hypertension OR: 1.200, 95% CI: 1.033–1.393; RERI: −0.18, 95% CI: −0.33 to −0.03; low HDL-C level OR: 1.163, 95% CI: 1.001–1.351; RERI: −0.16, 95% CI: −0.31 to −0.01). In women, the combination of low HGS and Western diet was associated with higher odds of IFG levels (OR: 1.223, 95% CI: 1.065–1.404; RERI: −0.22, 95% CI: −0.36 to −0.08), central obesity (OR: 1.453, 95% CI: 1.236–1.708; RERI: −0.42, 95% CI: −0.58 to −0.26), and low HDL-C levels (OR: 1.177, 95% CI: 0.995–1.393; RERI: −0.20, 95% CI: −0.37 to −0.04) (Figure 4). Results from unweighted models are presented in Appendix A.

## 4. Discussion

In this study, using KNHANES data from 2014 to 2019, low HGS was consistently associated with increased odds of MetS and its components, in line with previous findings. While dietary patterns alone were not significantly associated with MetS risk, their combination with low HGS was linked to increased odds and negative RERI values for several components, suggesting sub-additive effects.

Low HGS was associated with most components of metabolic syndrome, except for hypertension in females and IFG in males. These findings are consistent with previous studies conducted among adolescents and adults [3,26,27]. The lack of association between HGS and hypertension aligns with prior reports indicating that, while HGS is positively associated with hypertension in adults, such a relationship is not evident in adolescents [28,29]. Kalyani et al. reported that among the elderly (mean age 71.3 years), IFG was associated with persistently lower grip strength in older men but not in women, which contrasts with the sex-specific pattern observed in the present adolescent population [30].

In this study, dietary patterns derived from PCA were not independently associated with MetS or its components. This finding contrasts with previous studies that reported significant associations between unhealthy dietary patterns—such as high-fat or Western diets—and increased cardiometabolic risk in adolescents [7,8,14,15,16,17]. This discrepancy may be attributed to several factors, including methodological differences in how dietary patterns were assessed across studies. While some studies used predefined dietary indices such as the Mediterranean diet score or the Healthy Eating Index [31,32,33], our analysis utilized PCA to identify data-driven patterns, which may capture different aspects of dietary behavior. Although the Korean Healthy Eating Index has been developed to reflect national dietary guidelines, it was originally designed for adults and has not been validated for adolescents. Additionally, cultural and regional differences in food availability and eating habits—particularly among Korean adolescents—could influence the composition and health implications of each dietary pattern [34].

Beyond these methodological differences, traditional Eastern dietary patterns (e.g., kimchi, seaweeds) are rich in antioxidant and anti-inflammatory micronutrients—such as polyphenols, vitamins, minerals, and probiotics—and have been associated with lower Dietary Inflammatory Index scores [35]. These bioactive compounds improve insulin sensitivity and modulate lipid metabolism, thereby inhibiting the development of MetS [36]. Consequently, a shift toward a Western-style diet among Korean adolescents may reduce these protective factors and potentially increase their cardiometabolic risk.

However, when dietary patterns were examined in combination with HGS, a notable pattern emerged: adolescents with both low HGS and unhealthy dietary patterns—such as processed fat or Western-style diets—showed significantly increased odds of several components of MetS, particularly central obesity and low HDL-C. Importantly, these joint exposures were accompanied by negative RERI values, indicating sub-additive interactions. In other words, the combined effect of low HGS and poor dietary quality was smaller than the sum of their individual effects but still posed a meaningful metabolic burden.

While the negative RERI suggests a sub-additive interaction, the consistently elevated risk observed in individuals with both low HGS and poor dietary patterns may reflect an underlying physiological vulnerability. In this context, poor muscular strength may not statistically modify the effect of diet in a synergistic way, but may still identify a subgroup more susceptible to the adverse metabolic impact of unhealthy eating. Both low muscular strength and unhealthy diet were independently associated with increased MetS risk. Adolescents with low HGS may engage in lower levels of physical activity and have altered energy balance, further exacerbating the adverse effects of diets high in saturated fat and refined carbohydrates. Together, these factors may contribute to an impaired ability to compensate for poor dietary intake [37], resulting in a greater risk of metabolic dysregulation—even in the absence of additive interaction on a statistical scale.

Sex-specific differences were observed in the associations between HGS, dietary patterns, and metabolic syndrome components. In females, low HGS was significantly associated with IFG, whereas this association was not observed in males. These findings align with previous studies suggesting that adolescent girls may exhibit greater metabolic sensitivity to changes in insulin regulation, partly influenced by pubertal hormonal changes such as estrogen fluctuation [38]. During puberty, insulin resistance increases physiologically, with some studies reporting that girls exhibit a stronger postprandial insulinemic response than boys, particularly between the ages of 11 and 14 years. Moreover, the peak in insulin resistance tends to occur earlier in girls, typically around Tanner stage 3, which may have affected the results if some girls in our study population (aged 12–18 years) had already passed this stage [39].

The strengths of this study include the use of nationally representative data and the novel investigation of the interaction between low skeletal muscle mass, represented by HGS, and dietary patterns in relation to metabolic syndrome. Dietary patterns were identified using PCA, allowing a data-driven classification of dietary behaviors. The study also applied both weighted and unweighted analyses, considering the complex sampling design. Importantly, the analysis focused on adolescents—a critical life stage for the early prevention of chronic metabolic disorders. Furthermore, sex-stratified analyses provided insights into sex-specific associations, supporting the need for tailored public health interventions targeting modifiable lifestyle factors.

This study has several limitations. First, our analysis used cross-sectional, not longitudinal data. The results cannot reflect causality, only correlation; we cannot confirm direct or indirect pathways between handgrip strength, dietary patterns, and MetS, nor exclude reverse causality—although we hypothesized, based on prior longitudinal and mechanistic studies, that low muscle strength and unhealthy diet jointly increase metabolic risk. Second, due to the survey design, dietary information was obtained from a single 24 h recall only, precluding assessment of multi-day intake patterns and potentially failing to reflect habitual diet. Future studies employing multi-day dietary records could provide more reliable measures of usual diet and potentially reveal stronger associations between consumption patterns and metabolic risk. Third, the prevalence and the use of medications for diabetes mellitus, dyslipidemia, and hypertension, which may have introduced bias in defining metabolic components, were not surveyed for adolescent ages. In addition, the use of PCA to derive dietary patterns involves subjective decisions regarding component retention and labeling, potentially limiting reproducibility. Lastly, this study was conducted in Korean adolescents and may not be directly generalizable to countries with different cultural and dietary practices. Consistent HGS–MetS associations in European and North American cohorts suggest our findings may be generalizable. A prospective study incorporating validated measures of dietary quality, such as the Healthy Eating Index or the Korean Healthy Eating Index, specifically tailored for adolescents, is needed to better understand the causal relationship between dietary patterns and metabolic risk.

## 5. Conclusions

In this nationally representative sample of Korean adolescents, low HGS was independently associated with higher risk of metabolic syndrome and its components, and its combination with dietary patterns revealed sub-additive interaction effects—particularly for central obesity and low HDL-C—marking the first population-based evidence of hand-grip strength and dietary pattern interplay in youth. This sub-additive interplay indicates that both preserving muscle strength and following healthy dietary habits are independently crucial for MetS prevention. Integrating routine HGS screening with targeted resistance training and nutrition education may help mitigate metabolic risk in this population.

## Figures and Tables

**Figure 1 nutrients-17-02282-f001:**
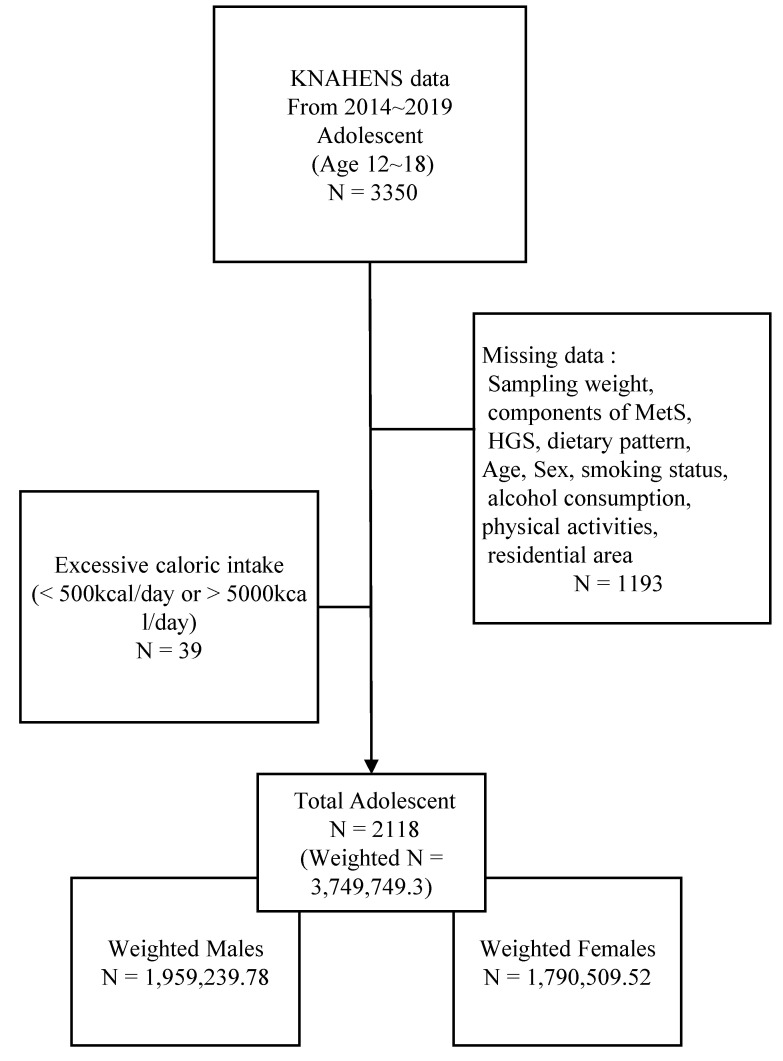
Study flowcharts. KNAHENS, Korea National Health and Nutrition Examination Survey; MetS, Metabolic syndrome; HGS, Handgrip strength.

**Figure 2 nutrients-17-02282-f002:**
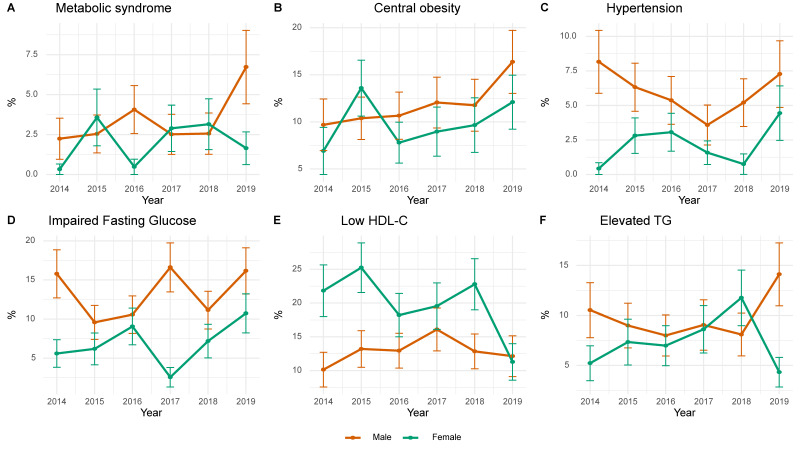
Annual trends in the prevalence of metabolic syndrome and its component conditions. HDL-C, High-density lipoprotein cholesterol; TG Triglyceride.

**Figure 3 nutrients-17-02282-f003:**
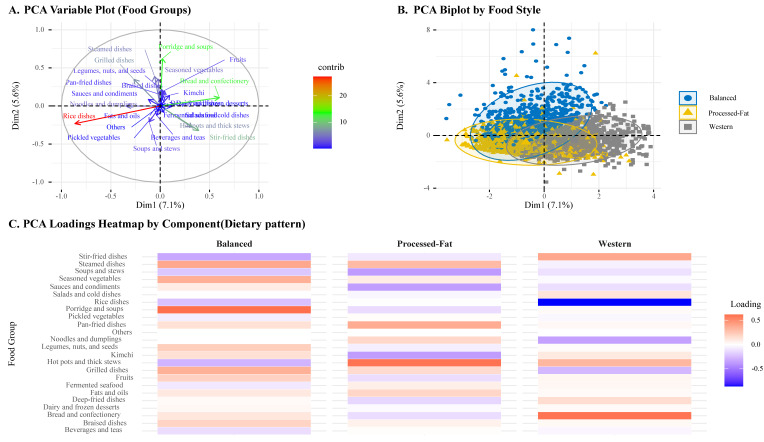
Principal Component Analysis of Dietary Patterns Based on Food-Group Intakes.

**Figure 4 nutrients-17-02282-f004:**
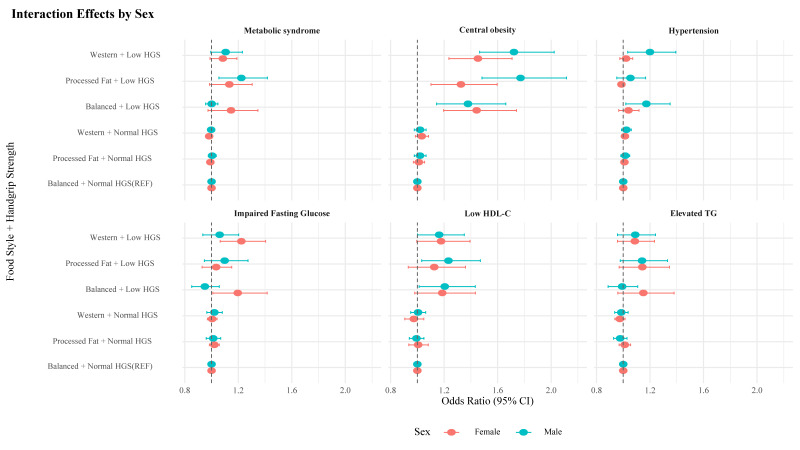
Sex-specific interaction effects of dietary pattern and handgrip strength on metabolic syndrome and its components. HGS, handgrip strength; HDL-C, high-density lipoprotein cholesterol; TG, triglyceride.

**Table 1 nutrients-17-02282-t001:** Characteristics of study populations.

			Unweighted (N, %)		Weighted (%)	
**Male**	**MetS**	**Normal**	** *p* **	**MetS**	**Normal**	** *p* **
	(*n* = 35)	(*n* = 1057)		(*n* = 61,791.83)	(*n* = 1,897,447.95)	
Age	16.0 ± 1.6	14.7 ± 2.0	<0.001	16.2 ± 0.2	15.2 ± 0.1	0.005
Dietary pattern			0.197			0.168
- Balanced	6 (17.1%)	327 (30.9%)		18.7	32.8	
- Processed Fat	16 (45.7%)	374 (35.4%)		46.1	37.5	
- Western	13 (37.1%)	356 (33.7%)		35.1	29.7	
Handgrip strength			<0.001			0.002
- Normal	19 (54.3%)	964 (91.2%)		63.9	91.2	
- Low	16 (45.7%)	93 (8.8%)		36.1	8.8	
Socioeconomic status			0.386			0.563
- High	15 (42.9%)	325 (30.7%)		37.4	30	
- Mid-high	8 (22.9%)	368 (34.8%)		22.7	33.9	
- Mid-low	9 (25.7%)	264 (25.0%)		28.1	25.5	
- Low	3 ( 8.6%)	100 (9.5%)		11.8	10.6	
Residential area			0.689			0.466
- Rural	22 (62.9%)	613 (58.0%)		62.2	55.5	
- Urban	13 (37.1%)	444 (42.0%)		37.8	44.5	
Smoking status			0.628			0.987
- No	29 (82.9%)	921 (87.1%)		84	84.1	
- Yes	6 (17.1%)	136 (12.9%)		16	15.9	
Alcohol consumption			1			0.971
- No	31 (88.6%)	940 (88.9%)		85.2	85.4	
- Yes	4 (11.4%)	117 (11.1%)		14.8	14.6	
Physical activity			1			0.650
- N	21 (60.0%)	624 (59.0%)		62.2	58.2	
- Y	14 (40.0%)	433 (41.0%)		37.8	41.8	
**Female**	**MetS**	**Normal**	** *p* **	**MetS**	**Normal**	** *p* **
	(*n* = 18)	(*n* = 1008)		(*n* = 29,581.48)	(*n* = 1,760,928.04)	
Age	16.1 ± 2.1	14.8 ± 2.0	0.007	16.4 ± 0.4	15.2 ± 0.1	0.005
Dietary pattern			0.482			0.288
- Balanced	7 (38.9%)	268 (26.6%)		47.5	26.7	
- Processed Fat	6 (33.3%)	366 (36.3%)		30.7	35.7	
- Western	5 (27.8%)	374 (37.1%)		21.8	37.6	
Handgrip strength			<0.001			0.003
- Normal	6 (33.3%)	919 (91.2%)		33.9	91	
- Low	12 (66.7%)	89 (8.8%)		66.1	9	
Socioeconomic status			0.074			0.018
- High	1 (5.6%)	350 (34.7%)		5.1	33.9	
- Mid-high	9 (50.0%)	331 (32.8%)		48.5	32.4	
- Mid-low	3 (16.7%)	103 (10.2%)		21.7	11.7	
- Low	5 (27.8%)	224 (22.2%)		24.7	22.1	
Residential area			0.772			0.914
- Rural	9 (50.0%)	567 (56.2%)		55.2	53.8	
- Urban	9 (50.0%)	441 (43.8%)		44.8	46.2	
Smoking status			1			0.860
- No	17 (94.4%)	955 (94.7%)		94.8	93.9	
- Yes	1 (5.6%)	53 (5.3%)		5.2	6.1	
Alcohol consumption			0.812			0.769
- No	16 (88.9%)	939 (93.2%)		89.3	91.4	
- Yes	2 (11.1%)	69 (6.8%)		10.7	8.6	
Physical activity			0.907			0.649
- N	12 (66.7%)	630 (62.5%)		69	63.4	
- Y	6 (33.3%)	378 (37.5%)		31	36.6	

MetS, metabolic syndrome.

**Table 2 nutrients-17-02282-t002:** Multivariable logistic regression stratified by sex.

		Male	Female
Metabolic Syndrome			
Handgrip Strength	Normal	1 (Ref)	1 (Ref)
	Low HGS	1.108 (1.038–1.182)	1.128 (1.047–1.216)
Food style	Balanced	1 (Ref)	1 (Ref)
	Processed Fat	1.024 (0.992–1.056)	0.982 (0.953–1.013)
	Western	1.006 (0.980–1.033)	0.975 (0.947–1.005)
Central obesity			
Handgrip Strength	Normal	1 (Ref)	1 (Ref)
	Low HGS	1.596 (1.438–1.771)	1.395 (1.258–1.547)
Food style	Balanced	1 (Ref)	1 (Ref)
	Processed Fat	1.039 (0.987–1.094)	0.977 (0.928–1.029)
	Western	1.036 (0.985–1.090)	1.023 (0.966–1.085)
Hypertension			
Handgrip Strength	Normal	1 (Ref)	1 (Ref)
	Low HGS	1.124 (1.040–1.215)	1.010 (0.978–1.044)
Food style	Balanced	1 (Ref)	1 (Ref)
	Processed Fat	1.002 (0.967–1.038)	1.003 (0.981–1.024)
	Western	1.021 (0.983–1.062)	1.009 (0.986–1.032)
Hyperglycemia			
Handgrip Strength	Normal	1 (Ref)	1 (Ref)
	Low HGS	1.024 (0.949–1.104)	1.156 (1.060–1.260)
Food style	Balanced	1 (Ref)	1 (Ref)
	Processed Fat	1.026 (0.974–1.080)	0.999 (0.958–1.041)
	Western	1.031 (0.977–1.087)	1.004 (0.962–1.048)
Low HDL-C			
Handgrip Strength	Normal	1 (Ref)	1 (Ref)
	Low HGS	1.201 (1.091–1.322)	1.177 (1.061–1.306)
Food style	Balanced	1 (Ref)	1 (Ref)
	Processed Fat	0.994 (0.940–1.051)	0.992 (0.925–1.065)
	Western	0.999 (0.945–1.056)	0.974 (0.907–1.045)
High TG			
Handgrip Strength	Normal	1 (Ref)	1 (Ref)
	Low HGS	1.089 (1.008–1.175)	1.127 (1.030–1.234)
Food style	Balanced	1 (Ref)	1 (Ref)
	Processed Fat	0.992 (0.944–1.042)	1.001 (0.955–1.050)
	Western	0.995 (0.948–1.045)	0.969 (0.927–1.013)

HGS, handgrip strength; HDL-C, high-density lipoprotein cholesterol; TG, triglyceride.

**Table 3 nutrients-17-02282-t003:** Interaction effects of handgrip strength and dietary patterns on metabolic syndrome and its components.

Outcome	HGS	Dietary Pattern	OR (Male) (95% CI)	RERI (Male) (95% CI)	OR (Female) (95% CI)	RERI (Female) (95% CI)
Metabolic Syndrome	Normal	Balanced	1.00 (Ref)	—	1.00 (Ref)	—
		Processed Fat	1.005 (0.976–1.036)	—	0.991 (0.971–1.011)	—
		Western	0.997 (0.971–1.023)	—	0.982 (0.964–1.000)	—
	Low	Balanced	1.001 (0.956–1.048)	—	1.146 (0.974–1.347)	—
		Processed Fat	1.223 (1.055–1.418)	−0.22 (−0.37–−0.07)	1.133 (0.985–1.304)	−0.14 (−0.28–0.00)
		Western	1.106 (0.994–1.231)	−0.11 (−0.21–0.00)	1.085 (0.989–1.190)	−0.10 (−0.19–−0.01)
Central Obesity	Normal	Balanced	1.00 (Ref)	—	1.00 (Ref)	—
		Processed Fat	1.020 (0.977–1.065)	—	1.011 (0.970–1.054)	—
		Western	1.020 (0.977–1.065)	—	1.033 (0.985–1.083)	—
	Low	Balanced	1.378 (1.144–1.661)	—	1.443 (1.196–1.741)	—
		Processed Fat	1.771 (1.483–2.115)	−0.75 (−0.93–−0.57)	1.326 (1.102–1.597)	−0.32 (−0.50–−0.13)
		Western	1.722 (1.465–2.023)	−0.70 (−0.86–−0.54)	1.453 (1.236–1.708)	−0.42 (−0.58–−0.26)
Hypertension	Normal	Balanced	1.00 (Ref)	—	1.00 (Ref)	—
		Processed Fat	1.015 (0.982–1.048)	—	1.009 (0.987–1.031)	—
		Western	1.023 (0.987–1.060)	—	1.013 (0.989–1.037)	—
	Low	Balanced	1.173 (1.019–1.350)	—	1.039 (0.966–1.118)	—
		Processed Fat	1.054 (0.952–1.167)	−0.04 (−0.14–0.06)	0.987 (0.970–1.004)	0.02 (0.00–0.04)
		Western	1.200 (1.033–1.393)	−0.18 (−0.33–−0.03)	1.021 (0.974–1.071)	−0.01 (−0.06–−0.04)
Hyperglycemia	Normal	Balanced	1.00 (Ref)	—	1.00 (Ref)	—
		Processed Fat	1.013 (0.960–1.069)	—	1.021 (0.984–1.058)	—
		Western	1.021 (0.965–1.081)	—	1.005 (0.969–1.042)	—
	Low	Balanced	0.950 (0.852–1.058)	—	1.196 (1.010–1.416)	—
		Processed Fat	1.098 (0.947–1.273)	−0.09 (−0.23–0.06)	1.035 (0.930–1.151)	−0.01 (−0.12–−0.09)
		Western	1.061 (0.935–1.203)	−0.04 (−0.17–0.09)	1.223 (1.065–1.404)	−0.22 (−0.36–−0.08)
Low HDL-C	Normal	Balanced	1.00 (Ref)	—	1.00 (Ref)	—
		Processed Fat	0.993 (0.940–1.049)	—	1.006 (0.936–1.082)	—
		Western	1.005 (0.950–1.062)	—	0.974 (0.906–1.048)	—
	Low	Balanced	1.205 (1.013–1.433)	—	1.186 (0.980–1.434)	—
		Processed Fat	1.232 (1.032–1.471)	−0.24 (−0.42–−0.06)	1.126 (0.932–1.360)	−0.12 (−0.30–0.07)
		Western	1.163 (1.001–1.351)	−0.16 (−0.31–−0.01)	1.177 (0.995–1.393)	−0.20 (−0.37–−0.04)
High TG	Normal	Balanced	1.00 (Ref)	—	1.00 (Ref)	—
		Processed Fat	0.977 (0.928–1.028)	—	1.011 (0.969–1.055)	—
		Western	0.985 (0.936–1.036)	—	0.975 (0.938–1.013)	—
	Low	Balanced	0.992 (0.887–1.108)	—	1.150 (0.959–1.380)	—
		Processed Fat	1.141 (0.978–1.331)	−0.16 (−0.32–−0.01)	1.143 (0.970–1.346)	−0.13 (−0.29–0.03)
		Western	1.090 (0.957–1.241)	−0.10 (−0.23–0.02)	1.087 (0.957–1.234)	−0.11 (−0.24–0.01)

OR, odds ratio; RERI, relative excess risk due to interaction; HDL-C, high-density lipoprotein cholesterol; TG, triglyceride.

## Data Availability

The datasets analyzed during the current study are available from the Korea National Health and Nutrition Examination Survey (KNHANES) website (https://knhanes.kdca.go.kr, accessed on 10 March 2025), managed by the Korea Disease Control and Prevention Agency (KDCA). Access to the data is granted upon request for research purposes in accordance with the data usage guidelines provided by KDCA.

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
