# Peer review of "Relative Excess Risk of Metabolic Syndrome Due to Interaction Between Handgrip Strength and Dietary Patterns Among Korean Youth"

_nutrients, 2025, doi:10.3390/nu17142282_

Round 1

Reviewer 1 Report

Comments and Suggestions for Authors

Yoon et al compared here how indexes of muscle strength (hand-grip strength, in fact) would affect the correlations between inappropriate dietary habits and incidence of Metabolic Syndrome in Korean adolescents.

The MS was well-written and clearly brought up the scientific question involved in the study, which was developed through a robust screening (multivariate logistic regression analysis) from KNHANES data, from 2014 to 2019, starting with 3.75M individuals (reaching a final sample of +2,000 adolescents, 12-18 yo).

I consider this MS a very solid contribution to the field, but, from my point of view, there are still few issues that have to be answered/corrected before its acceptance in Nutrients/MDPI:

(i) Table 3 is confusing because of its format. Please reorganize columns width, font sizes etc.

(ii) I missed a Conclusion heading in the MS, in order to summarize the main findings of the study in a couple of sentences.

(iii) Please, include one or two paragraphs to discuss the potential role of micronutrients present in the traditional Eastern dietary (long-recognized by its healthy components) - presumably depleted by acquiring western dietary habits - to prevent MS in adulthood associated or not with physical activity.

Author Response

(i) Table 3 is confusing because of its format. Please reorganize columns width, font sizes etc.

  • We Revised the Table3 to enhance visibility and readability

(ii) I missed a Conclusion heading in the MS, in order to summarize the main findings of the study in a couple of sentences.

  • We added a new Conclusions section at the end of the manuscript, which briefly summarizes the key findings in two sentences:

(iii) Please, include one or two paragraphs to discuss the potential role of micronutrients present in the traditional Eastern dietary (long-recognized by its healthy components) - presumably depleted by acquiring western dietary habits - to prevent MS in adulthood associated or not with physical activity.

  • We inserted a new paragraph in the discussion, outlining how traditional Eastern dietary components (kimchi, seaweeds, green vegetables, legumes) supply polyphenols, vitamins, minerals, and probiotics that lower dietary inflammatory index scores, improve insulin sensitivity, and modulate lipid metabolism—thereby protecting against metabolic syndrome.

“Beyond these methodological differences, traditional Eastern dietary patterns (e.g., kimchi, seaweeds) are rich in antioxidant and anti-inflammatory micronutrients—such as polyphenols, vitamins, minerals, and probiotics—and have been associated with lower Dietary Inflammatory Index scores [35]. These bioactive compounds improve insulin sensitivity and modulate lipid metabolism, thereby inhibiting the development of MetS [36]. Consequently, a shift toward a Western-style diet among Korean adolescents may reduce these protective factors and potentially increase their cardiometabolic risk”

Reviewer 2 Report

Comments and Suggestions for Authors

Dear Authors,
Could you please clarify the following:
1. Do all the studies reflect the same relationship between hand grip strength (HGS) and metabolic syndrome (MetS)? Is it linear and does it take into account the influence of other factors? Please expand on this relationship in the introduction.

2. Is HGS directly related to dietary factors and indirectly related to MetS?

3. Does the use of the 24-hour dietary interview method not fully reflect the subjects' diet? Perhaps a 3- or 7-day dietary record should have been used instead. Then the authors would have achieved more significant results regarding the relationship between consumption and other factors.

4. Please justify the inclusion of alcoholic beverages in the study group (12–18 years old). From what age is the sale of alcohol permitted in Korea?

5. Do the study's conclusions contribute significantly to current knowledge about metabolic syndrome among young people?

6. What are the implications of the results for health policy or prevention programmes?

7. Are there any indications that the results can be generalised to other populations (outside of Korea)?

Best regards,

Reviewer

Author Response

  1. Do all the studies reflect the same relationship between hand grip strength (HGS) and metabolic syndrome (MetS)? Is it linear and does it take into account the influence of other factors? Please expand on this relationship in the introduction.
    • We have expanded the Introduction as follows:

“In Korean adolescents aged 10–18, Jung et al. (3) reported a significant P-for-trend for normalized combined HGS (HGS/body weight) with both metabolic syndrome and insulin resistance after adjusting for sex, age, physical activity, and sedentary time. Wen et al. (5) demonstrated a linear dose–response relationship—each 0.1-unit decrease in relative HGS (HGS/body weight) was associated with a 32% higher risk of MetS after adjustment for age, sex, BMI, socioeconomic status, smoking, and physical activity.”

  1. Is HGS directly related to dietary factors and indirectly related to MetS?
    • As our analysis used cross-sectional data, we cannot infer causality or distinguish direct from indirect pathways between handgrip strength, dietary patterns, and MetS, nor rule out reverse causality. However, based on prior longitudinal and mechanistic studies, we hypothesized that low muscle strength and unhealthy diet jointly increase metabolic risk.

“First, our analysis used cross-sectional, not longitudinal data. The results cannot reflect causality, only correlation; we cannot confirm direct or indirect pathways between handgrip strength, dietary patterns, and MetS, nor exclude reverse causality—although we hypothesized, based on prior longitudinal and mechanistic studies, that low muscle strength and unhealthy diet jointly increase metabolic risk.”

  1. Does the use of the 24-hour dietary interview method fully reflect the subjects’ habitual diet? Perhaps a 3- or 7-day dietary record would have yielded stronger consumption–risk associations.
    • KNHANES collects only single 24-hour recalls and does not include multi-day records. We have addressed this in the Limitations:

“Second, due to the survey design, dietary information was obtained from a single 24-hour recall only, precluding assessment of multi-day intake patterns and potentially failing to reflect habitual diet. Future studies employing multi-day dietary records could provide more reliable measures of usual diet and potentially reveal stronger associations between consumption patterns and metabolic risk.”

  1. Please justify the inclusion of alcoholic beverages in the study group (12–18 years old). From what age is the sale of alcohol permitted in Korea?
    • Although Korea’s legal drinking and smoking age is ≥19 years, national surveys report that adolescents sometimes smoke and consume alcohol. Therefore, we included both smoking status and alcohol consumption as covariates.

“Although the legal smoking and drinking age in Korea is ≥19 years, smoking experience and occasional alcohol use (≥1 drink/month) are reported among adolescents; therefore, we included smoking status and alcohol consumption as covariates (20).”

  1. Do the study’s conclusions contribute significantly to current knowledge about metabolic syndrome among young people?
  2. What are the implications of the results for health policy or prevention programmes?
    • To address these, we revised our Conclusions to emphasize that this is the first population-based evidence of HGS and dietary-pattern interplay in youth, and to recommend both muscle-strengthening and nutrition education interventions:

“In this nationally representative sample of Korean adolescents, low HGS was independently associated with higher risk of metabolic syndrome and its components, and its combination with dietary patterns revealed sub-additive interaction effects—particularly for central obesity and low HDL-C—marking the first population-based evidence of hand-grip strength and dietary pattern interplay in youth. This sub-additive interplay indicates that both preserving muscle strength and following healthy dietary habits are independently crucial for MetS prevention. Integrating routine HGS screening with targeted resistance-training and nutrition education may help mitigate metabolic risk in this population.”

  1. Are there any indications that the results can be generalised to other populations (outside of Korea)?
    • Our study was restricted to Korean adolescents, so direct generalization to other cultural or dietary contexts is limited. However, consistent HGS–MetS associations in European and North American cohorts suggest our findings may be applicable to similar populations.